# Teaching Smaller Language Models To Generalise To Unseen Compositional Questions

**Tim Hartill**                                                      *thar011@aucklanduni.ac.nz*
*School of Computer Science*
*University of Auckland*

**Neset TAN**                                                       *ntan607@aucklanduni.ac.nz*
*School of Computer Science*
*University of Auckland*

**Michael Witbrock**                                                *m.witbrock@auckland.ac.nz*
*School of Computer Science*
*University of Auckland*

**Patricia J. Riddle**                                              *p.riddle@auckland.ac.nz*
*School of Computer Science*
*University of Auckland*

**Reviewed on OpenReview:** *https://openreview.net/forum?id=d4Vr6E0jjm*

## Abstract

We equip a smaller Language Model to generalise to answering challenging compositional questions that have not been seen in training. To do so we propose a combination of multi-task supervised pretraining on up to 93 tasks designed to instill diverse reasoning abilities, and a dense retrieval system that aims to retrieve a set of evidential paragraph fragments. Recent progress in question-answering has been achieved either through prompting methods against very large pretrained Language Models in zero or few-shot fashion, or by fine-tuning smaller models, sometimes in conjunction with information retrieval. We focus on the less explored question of the extent to which zero-shot generalisation can be enabled in smaller models with retrieval against a corpus within which sufficient information to answer a particular question may not exist. We establish strong baselines in this setting for diverse evaluation datasets (StrategyQA, CommonsenseQA, IIRC, DROP, Musique and ARC-DA), and show that performance can be significantly improved by adding retrieval-augmented training datasets which are designed to expose our models to a variety of heuristic reasoning strategies such as weighing partial evidence or ignoring an irrelevant context.

## 1 Introduction

We are inspired by recent progress with pretrained large Language Models (LLM), which when prompted with task demonstrations (Brown et al., 2020), instructions (Sanh et al., 2021; Wei et al., 2021; Ouyang et al., 2022) or reasoning chains (Wei et al., 2022), show an ability to answer questions unlikely to have been encountered during training. However a diversity of potential applications exist in the broad domain of reasoning systems and considerations such as latency, cost, energy efficiency, physical compute size and internet connectivity requirements are relevant in determining the most appropriate approach for a given situation.

Rather than encoding all knowledge in the parameters of a LLM, an alternative approach has been to transform the original question-answering problem into a reading comprehension (RC) problem by retrieving

relevant information for answering a particular query from an external corpus, and training a smaller[1] model (QA Model) to reason over the concatenation of the query and retrieved information to derive an answer e.g. Chen et al. (2017). In this paper[2] we extend retrieval methods as described in section 1.1 in conjunction with a supervised multitask pretraining regime for the QA Model involving 79 tasks for our baseline and 93 tasks for the improved model.

The viability of this approach outside of fine-tuned settings is currently subject to limitations, both in the retrieval component, as discussed below, and with respect to the inabilities of smaller language models to perform the reasoning function as well as larger models. We aim to quantify performance limitations and evaluate mitigations for some of them.

There are at least two significant challenges in retrieval to be overcome. Firstly, no matter how large the corpus is, there will always be missing information, particularly so in our setting where neither datasets nor corpus have been normalised such that sufficient information is in the corpus to make each question answerable through deductively valid means. Secondly, as long as humans ask questions with ambiguous references e.g. "Who is the spouse of the Green performer?" (Trivedi et al., 2022a), retrieval will necessarily be imperfect even where sufficient knowledge exists in the corpus and the retrieval method is otherwise perfect.

We evaluate a method for addressing these issues. Specifically, we measure the effect of adding datasets to our QA Model training regime that are designed to impart heuristic strategies for reasoning to a plausible rather than an entailed answer. We construct these datasets by building contexts for training questions using our retrieval system against a fixed corpus of English Wikipedia paragraphs. The resulting samples ("retrieval-augmented training datasets", abbreviated to *RATD*) are included in training our QA Model irrespective of whether they contain partial, full, or no evidence. Our approach carries the advantage that a diversity of reasoning strategies may be imparted. Such strategies include ignoring an irrelevant context completely or weighing partially evidential facts; e.g. reasoning toward answering "Do teenagers always rebel against their parents?" (Talmor et al., 2021) can be aided by the retrieval of knowledge that "Adolescents who have a good relationship with their parents are less likely to engage in various risk behaviours", even though there is no entailment implied.

Generally our method is most applicable to question-answering tasks where the desired answer is short i.e. from a word to a short sentence, and the question itself does not come already supplied with a fully evidential context. We also assume that it is possible to retrieve sufficient information from our corpus so as to make a question answerable within a modest sequence length (we limit ourselves to a 512 token maximum) e.g. we are unlikely to be able to answer a question such as "How many songs have a person's name in the title?" even through retrieving every instance is theoretically possible. We focus our study on a set of unseen evaluation datasets that meet the following criteria: (1) Datasets collectively involve diverse textual and numerical reasoning strategies. (2) Questions are generally readily answerable by humans with access to a web browser and without specialised knowledge. (3) Questions tend to be compositional. (4) Relevant comparison with prior work exists. In regards to defining compositionality, others e.g. (Dankers et al., 2022; Hupkes et al., 2020) have noted challenges in singularly defining compositionality as it relates to NLP; for our purposes we pragmatically define a question as compositional if it is unlikely to be answerable by our QA Model with a memorised answer from a similar training example, and requires reasoning over a context by utilising at least one reasoning operation (e.g. conjunction) involving more than one textual fact, and/or at least one numerical operation involving more than one number.

This criteria leads us to select six evaluation datasets: StrategyQA (Geva et al., 2021) contains commonsense samples requiring diverse multi-hop reasoning strategies. On average samples require content from 2.33 separate paragraphs to answer when considering retrieval from Wikipedia. Musique (Trivedi et al., 2022a) is a multi-hop dataset focused on factual questions that require retrieval of content from two to four paragraphs. IIRC (Ferguson et al., 2020) contains questions where an initial paragraph is given and answers depend upon reasoning over this plus one to over four additional paragraphs that must be retrieved. ARC-DA

---

[1]We define smaller language models as generative Transformer models with 400 million to 1 billion parameters, i.e those that are large enough to be effective reasoners whilst being able to perform training and inference with reasonable latency, cost and energy efficiency.

[2]Code, models and datasets will be released at `https://github.com/timhartill/unseen_questions`

(Bhakthavatsalam et al., 2021) is a question-only subset of ARC (Clark et al., 2018). The Worldtree database provides explanatory fact sets for ARC samples which average six facts per sample (Xie et al., 2020). DROP (Dua et al., 2019) is a RC dataset wherein answering each question requires numerical or temporal reasoning over a provided context to reach an often abstractive answer e.g. "How many field goals were scored in the first quarter? ...The Lions scored first...with a 23-yard field goal...The Buccaneers tied it up with a 38-yard field goal...then took the lead...The Lions responded with a 28-yard field goal..." The answer is 3 which isn't explicit in the context. CommonsenseQA (Talmor et al., 2019) contains samples that are often unlikely to be answerable by finding a singular fact e.g. "I'm crossing the river, my feet are wet but my body is dry, where am I? (A) waterfall (B) bridge (C) valley (D) bank (E) island" is answered by considering knowledge related to each option. These datasets are discussed more fully in section 3.1.

In addition to the possibility of answer leakage from directly memorised samples, it has been shown that models are able to utilise more subtle cues such as the writing style of a particular annotator who contributed to both train and test splits for better results than are achievable where the test split is truly independent of the training split (Geva et al., 2019). To minimise such issues as well as to facilitate comparison in a similar setting as other zero/few shot studies which have varying definitions of "seen-ness", we simply define an unseen question as one from an evaluation dataset that is disjoint from our training datasets. Against this definition, two of our evaluation datasets, ARC-DA and Musique, are "partially seen" as discussed further below.

In summary the major contributions of this paper are: (A) We offer what is to our knowledge the most comprehensive set of baselines evaluating smaller Language Model zero-shot reasoning abilities published to date. (B) We show that augmenting the training regime with *RATD* datasets significantly improves performance from the baselines. (C) We demonstrate that training for numerical literacy and unanswerability is brittle in the unseen setting in the absence of sufficiently similarly formatted training examples. (D) We propose effective extensions to the retrieval approach as described below.

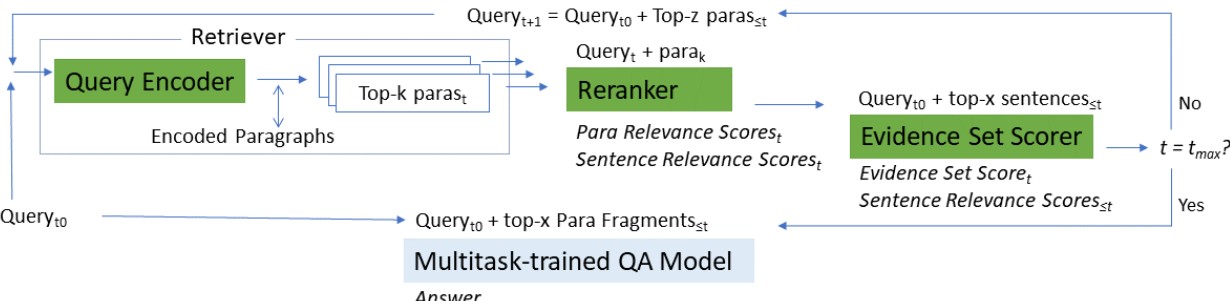

Figure 1: Major system components: The Iterator (green boxes) and QA Model. An initial query for hop $t=0$ is input into the Retriever. The Reranker scores each of the retrieved $k$ paragraphs and constituent sentences. Top-$x$ sentences (Evidence Set$_{\leq t}$) are selected from top-ranked sentences from the Reranker and from the prior hop Evidence Set$_{<t}$. The query + Evidence Set$_{\leq t}$ are input into the Evidence Set Scorer which computes an overall Evidence Set Relevance Score $e$ and individual sentence relevance scores. Paragraphs associated with the top five sentences of Evidence Set$_{\leq t}$ are appended to the query and the process repeats $t_{max}$ times. Finally, paragraph fragments recovered from the Evidence Set for hop $t=\arg\max(e)$ are concatenated with the original query and input into the QA Model for answer generation.

## 1.1   System Components and Related Work

**Retrieval**. Chen et al. (2017) first used sparse retrieval, namely TF-IDF (Spärck Jones, 1972), against Wikipedia in the context of open domain question-answering. In dense retrieval, query and corpus documents are embedded into the same vector space and retrieval is typically performed through maximum inner product search (MIPS) over the resulting dense vectors. Several such approaches e.g. Karpukhin et al. (2020) focus on retrieving the single most relevant document sufficient for answering a single-hop query. Xiong et al. (2021) introduce multi-hop dense retrieval (MDR), to retrieve *multiple* documents necessary to answer a

complex multi-hop question. They focus on the two-hop situation where a maximum of two documents are sufficient to answer a question. In this situation training samples are input to a shared question and document encoder as: (1) Input $\langle q_i \rangle$ with an objective of minimizing distance to the vector representing $d_{i,0}$ (hereafter denoted $\langle q_i \rangle \to d_{i,0}$), where $q_i$ and $d_{i,t}$ are the *i-th* question and the *t-th* supporting document of $q_i$ to retrieve respectively. (2) Input $\langle q_i, d_{i,0} \rangle \to d_{i,1}$. We extend the MDR training regime and loss computation to enable retrieval of an arbitrary maximum number of documents i.e. $\langle q_i, d_{i,0}, ..., d_{i,t} \rangle \to d_{i,t+1}$.

Wang et al. (2018) introduced the concept of a Reranker that refines retrieved results. IRRR (Qi et al., 2021) combined sparse retrieval and reranking into an iterative single model that can also answer multi-hop questions that have extractive answers. Baleen (Khattab et al., 2021), is also iterative but uses a dense retrieval system based upon encoding a dense vector per input token. Their two-stage condenser system comprises a Reranker that scores the relevance of each sentence for each retrieved document followed by an additional module that scores relevance of each sentence from the top-scoring sentences selected over multiple documents from the first stage. It then generates a compressed context of relevant sentences, to be utilised by a separate QA Model. We take inspiration from Baleen's two-stage approach but other than using our own retriever, we differ most notably in that we introduce an Evidence Set Score into the second stage with the goal of quantifying the sufficiency of the entire set of selected sentences for answering a query, in addition to scoring the relevance of individual sentences.

Sparse retrieval offers the advantage that it can perform well in zero-shot settings where lexical overlap is sufficient to identify relevant documents. Several studies evaluate methods that improve the performance of dense retrieval models in zero-shot settings. A number of these use diverse unsupervised techniques involving creating queries and positive passages from unlabelled text e.g. (Lee et al., 2019; Ram et al., 2022; Izacard et al., 2022). In a different approach, Chen et al. (2021) trained a dense retriever to imitate a lexical-based model with good results. Thakur et al. (2021) created the BEIR benchmark to further the study of retrieval in the zero-shot setting and a number of more recent papers report results against this benchmark. We are unable to do so as some of our retriever training datasets are BEIR components, however we note as a future direction that our retriever training might benefit further from applying techniques that have been effective on BEIR.

**Multitask Pretraining**. Raffel et al. (2020) showed that when trained using self-supervised pretraining followed by supervised multitask training, a single text-to-text Transformer model without task-specific architectural modification was capable of performing well on all the diverse tasks it had been trained upon. Since then, a number of studies have shown the efficacy of supervised multitask training in facilitating generalisation in question-answering tasks (Khashabi et al., 2020b; Sanh et al., 2021; Wei et al., 2021; Khashabi et al., 2022). Orthogonal to our approach, many studies e.g. Sanh et al. (2021); Wei et al. (2021); Ouyang et al. (2022) make use of instruction-based tuning to facilitate generalisation. In order to focus on evaluation of differing training data regimes, we make use of a similar fixed prompting format to Khashabi et al. (2020b; 2022) and utilise many of their converted QA datasets. Perhaps most similar to our work, Wang et al. (2022b) combines multitask training over multi-choice datasets with external retrieval which they use to augment the training set. However their implementation diverges from ours in that they use sparse retrieval and then a fusion-based method similar to Izacard & Grave (2021) wherein multiple retrieved document vectors are used with gated cross-attention to focus on salient information. Their evaluation datasets are disjoint with ours and don't cover broader reasoning skills like numeracy, so comparison must be left to future work.

Longpre et al. (2021) created a synthetic dataset by substituting entity names in existing dataset contexts and updating corresponding labels to produce new unfactual but logically consistent samples. They show that training on the new dataset plus the original causes their model to rely on reasoning over the context more, and less on knowledge encoded in parameters. Recently, Li et al. (2022) extended this approach to a fine-tuning framework for LLMs wherein the model is trained on relevant, irrelevant, and counterfactual but logically consistent contexts. Their approach differs from ours in that our *RATD* datasets are constructed so as to encourage reasoning to a plausible conclusion whereas theirs are constructed with logical entailment in mind i.e. to ignore contexts where deductively valid reasoning is not possible in favor of knowledge stored in the LLM parameters.

**Numerical Literacy**. Yoran et al. (2022), Pi et al. (2022) and Geva et al. (2020) all develop numeracy-focused pretraining datasets that we adapt and utilise. Generally these approaches have concentrated on finetuned settings and to our knowledge we are the first to study their performance against a diversity of unseen evaluation datasets. Recently Trivedi et al. (2022b) released numeracy-focused pre-training datasets constructed using QDMR (Wolfson et al., 2020) decompositions. These were released too late for us to include in our pretraining regime but we report comparisons in Table 2.

## 2  Method

We develop and train the Retrieval, Reranking, Evidence Set Scoring (collectively the "Iterator"), and QA model components separately as visualised in Figure 1. Comparisons with retrieval systems in our setting are limited since gold paragraph annotation does not exist. Moreover, excepting Khashabi et al. (2020b; 2022) papers tend not to report zero-shot results for smaller language models such as the BART (Lewis et al., 2020) QA model we use. Therefore we initially evaluate the performance of components on in-domain settings with comparisons to strong prior work, and report results in this section. In subsequent sections we move to the major focus of our study, namely to evaluate our method of adding $RATD$ datasets to improve reasoning in the setting where questions are unseen, sufficient evidence to deductively answer a query may not be retrievable, and the model is too small to effectively answer open domain questions without a context to reason over.

### 2.1  Retrieval

For the retrieval component of the Iterator, as discussed above we extend MDR from a two hop maximum to enable training on samples with an arbitrary maximum number of hops ($t_{\max}$). Training is over a mixture of datasets with questions involving one to four hops to answer; HotpotQA (Yang et al., 2018), Hover (Jiang et al., 2020), Natural Questions (Kwiatkowski et al., 2019), and Musique (Trivedi et al., 2022a). Hence in practice we set $t_{\max} = 4$. Multi-hop questions contain multiple possible reasoning paths through the labelled gold paragraphs, some of which the encoder is able to learn to generalise from ("learnable") and some not (Xiong et al., 2021). For example, given a set of supporting documents for a 4-hop $q_i$ as $\{d_{i,0}, d_{i,1}, d_{i,2}, d_{i,3}\}$, semantic overlaps between $q_i$ and the documents might enable learnable reasoning paths of $\langle q_i, d_{i,0}, d_{i,1}, d_{i,2}, d_{i,3}\rangle$ or $\langle q_i, d_{i,1}, d_{i,0}, d_{i,3}, d_{i,2}\rangle$ but not $\langle q_i, d_{i,2}, d_{i,0}, d_{i,1}, d_{i,3}\rangle$ or others. Our training regime samples a learnable reasoning path and builds training samples for subsets; e.g. from $\langle q_i, d_{i,1}, d_{i,0}, d_{i,3}, d_{i,2}\rangle$ we would build four single-hop samples $\langle q_i\rangle \rightarrow d_{i,1}$, $\langle q_i, d_{i,1}\rangle \rightarrow d_{i,0}$, $\langle q_i, d_{i,1}, d_{i,0}\rangle \rightarrow d_{i,3}$ and $\langle q_i, d_{i,1}, d_{i,0}, d_{i,3}\rangle \rightarrow d_{i,2}$. We based document sequencing for learnable reasoning paths for Musique on the decomposed reasoning steps provided with that dataset. For HotpotQA and Hover we used the ordering that was used in Xiong et al. (2021) and Khattab et al. (2021) respectively, while Natural Questions is treated as single-hop.

For each training sample, positive documents from other training examples in the batch are used as negatives, to which are added two adversarially negative paragraphs specific to that question. Where adversarial negative documents were not otherwise available we created them from our Wikipedia corpus by taking the first paragraph of directly hyperlinked documents from each gold paragraph. Specifically, we used this strategy to create negative documents for Hover as well as to create additional negatives for Musique. We used adversarial negatives for HotpotQA and Natural Questions supplied from Xiong et al. (2021) and Karpukhin et al. (2020) respectively.

Our objective function is similar to others e.g. (Xiong et al., 2021; Karpukhin et al., 2020). For hop $t$ of the $i - th$ training sample it models the probability of each next document given a query as:

$$P(dvec_{i,t+1}|qvec_{i,t}) = \frac{exp(dvec_{i,t+1} \cdot qvec_{i,t})}{\sum_{dvec\in D_i} exp(dvec \cdot qvec_{i,t})}$$

Where $qvec_{i,t} = enc(\langle q_i, d_{i,0}, ..., d_{i,t}\rangle)$, $dvec_{i,t+1} = enc(\langle d_{i,t+1}\rangle)$, $enc$ is the shared encoder, $qvec_{i,t}$ is the encoded query vector, $dvec_{i,t+1}$ is the encoded next document vector, $D_i$ is the set of positive and negative document vectors for $q_i$ and $\cdot$ denotes the inner product operation.

## 2.2 Reranking and Evidence Set Scoring

To refine retrieved documents we implement a two-stage system comprising Reranker and Evidence Set Scoring models. Both models were trained using a mixture of datasets that come with sentence-level annotations, namely HotpotQA, Hover and FEVER (Thorne et al., 2018).

Training samples for the Reranker are built from learnable reasoning paths. For two-hop paths, samples are randomly built to one or two hops i.e. $\langle q_i, d_{i,0} \rangle$ to score $d_{i,0}$ relevance, or $\langle q_i, d_{i,0}, d_{i,1} \rangle$ to score $d_{i,1}$. To remediate imbalance in hop distribution three and four hop samples are always built to the respective maximum hop count. Each query is paired with both a positive paragraph to be scored, and a substituted negative paragraph. The sampling function implements a form of shared normalization (Clark & Gardner, 2018) such that pairs are positioned in the same training batch.

In the Reranker, a paragraph relevance score ($p$) in addition to individual sentence relevance scores ($s_p$) are learned. The objective function for each is binary cross-entropy with the overall loss being an unweighted summation. Intuitively, a high-scoring sentence in a relevant paragraph is more likely to be evidential than a high scoring sentence in an irrelevant paragraph. We manually observed that $p$ is often more accurate than $s_p$ and hence experimented with tuning a weight, $w$, in a sentence scoring function $s = wp + (1 - w)s_p$. For in-domain datasets such as HotpotQA we found non-zero values of $w$ that improved both sentence and paragraph recall by over 2%, and F1 score by over 6%, confirming our observation. However the optimal value of $w$ varied between 0.0 and 0.9 over in-domain datasets and tuning $w$ for any of our unseen datasets using their gold annotations would compromise our experimental setup. Hence we simply score each sentence in our main experiments as $s = 0.5p + 0.5s_p$.

For the second stage Evidence Set Scorer, at each hop $t$ the Evidence Set$_{\leq t}$ is selected from top-ranked sentences from the Reranker and from the prior Evidence Set$_{<t}$, if any. The query and Evidence Set$_{\leq t}$ are input into the Evidence Set Scorer which scores evidence set relevance ($e$), and sentence relevance ($s_e$) in the context of the evidence set. The sentences for the $t + 1$ evidence set are selected by ranking according to $0.5p + 0.5s_e$ and then taking a maximum of five sentences that score over a threshold. The 0.5 coefficients were chosen after a similar evaluation as was done for the Reranker scoring function described above. We observed instances where the evidence set weakened as well as where it strengthened with additional hops, so we then take the evidence set from hop $t = \arg\max(e)$ rather than assuming that $t_{max}$ always selects the best.

We observed that a high-scoring sentence is sometimes contextualized by adjacent sentences and collectively they create a stronger rationale. Hence final context for each query, both for *RATD* dataset creation and for creating context for unseen evaluation samples, is created by recovering a paragraph fragment for each selected sentence by prepending/appending the preceding and subsequent sentence from the associated full paragraph where these exist, and then concatenating the document title with the resulting fragment. Ordering of paragraph fragments is by $0.5p + 0.5s_{max}$ where $s_{max}$ is the maximum Evidence Set Scorer sentence relevance score per paragraph. Using these paragraph fragments it is possible to fit contexts of approximately 6-7 paragraph fragments within a 512-token maximum sequence length. In the case of datasets such as IIRC (Ferguson et al., 2020) that provide an initial paragraph in addition to the question, the initial paragraph is prepended to the context.

The Evidence Set Scoring model is trained with Evidence Sets built as combinations of positive and negative sentences, including replacing positive sentences with negative sentences from positive paragraphs and negative sentences from negative paragraphs. Each question is paired with both a fully evidential set of sentences and a partially evidential (or non-evidential) set of sentences sampled such that pairs are in the same training batch. The objective functions for both $e$ and $s_e$ are binary cross-entropy and as with the Reranker the final loss is an unweighted summation. The label for $e$ is 1.0 if a subset of the Evidence Set is fully evidential, 0.0 otherwise.

Further details of the Iterator components are in Appendix D.

Table 1: In-domain Retrieval and Reranking Evaluation on Hover development set with $k = 25$. Baleen is finetuned on Hover, MDR is trained on HotpotQA, and our retriever is trained on a mixture of HotpotQA, Hover, Musique and Natural Questions.

| Model / # of Hops–> | Sentence EM | | | | Sentence F1 | | | |
|---|---|---|---|---|---|---|---|---|
| | **2** | **3** | **4** | **All** | **2** | **3** | **4** | **All** |
| Baleen 4-hop + FLIPR retriever | 47.3 | 37.7 | **33.3** | 39.2 | 81.2 | **82.5** | **80.0** | **81.5** |
| Iterator + MDR retriever | 64.6 | 39.3 | 14.8 | 40.1 | 81.7 | 72.1 | 59.0 | 71.4 |
| Iterator + our retriever | **66.7** | **45.4** | 27.5 | **46.8** | **82.5** | 75.7 | 68.7 | 75.8 |

### 2.3 Iterator In-domain Evaluation

We initially evaluate performance of the Iterator in an in-domain setting using the Hover development set against the HotpotQA Wikipedia Abstracts Corpus (Yang et al., 2018), since Hover contains samples with up to four hops and it is possible to compare against the published Baleen performance. Here the number of paragraphs retrieved on each hop ($k$) is 25. Results (Table 1) indicate that our Iterator is competitive with Baleen in this setting with our two-hop performance better but their four-hop performance dominating. A reason we are stronger overall than Baleen on EM while the reverse is true for F1 is due to our choice of ranking function - Baleen ranks sentences entirely using $s_e$ whereas we utilise a linear combination of our Reranker paragraph score $p$ and $s_e$. Unsurprisingly our retriever performance is progressively better than MDR as the number of hops increases.

Our main experiments below use a corpus consisting of English Wikipedia paragraphs from the August 1 2020 dump. Details are in Appendix C.

### 2.4 QA Models

A number of studies have shown the efficacy of supervised multitask training in facilitating generalisation in question-answering tasks (Khashabi et al., 2020b; Sanh et al., 2021; Wei et al., 2021; Khashabi et al., 2022). We adopt this approach for training our QA models.

To facilitate numerical computation we adapt the QA model tokenizer for digit tokenisation (Wallace et al., 2019; Geva et al., 2020) in all experiments.

Noting that some of the numerical pretraining tasks take much longer to train to a reasonable degree of proficiency than our textual question-answering tasks, we continue training our QA models from their original pretraining checkpoint with two additional stages of multitask pretraining.

#### 2.4.1 Stage 1 Pretraining

In Stage 1 we train using tasks that are aimed at imparting by abstraction a diversity of foundational reasoning skills, with a bias towards simple numerical literacy. Specifically we utilise existing tasks from Yoran et al. (2022), Pi et al. (2022) and Geva et al. (2020) as well as some we create ourselves. Stage 1 training is on a total of 33 tasks. One of these is a version of the original self-supervised masked language modelling task which is sampled with probability $\lambda = 0.35$. The remaining tasks are sampled using an error-based sampling regime (Gottumukkala et al., 2020) whereby tasks with low accuracy in the previous validation step are oversampled in the subsequent training steps and vice-versa.

#### 2.4.2 Stage 2 Pretraining

In Stage 2, we add five open domain (i.e. question-only) question-answering tasks to the above foundational Stage 1 tasks (for 38 tasks in total, denoted *Group 1*). We add the open domain tasks with the primary aim of teaching the model about the expected form of answer for a given question type e.g. yes or no for "Could an Aardvark use a knife and fork?" noting that it has been shown that smaller models cannot learn such open domain tasks well (Lewis et al., 2021). To avoid the possibility of catastrophic forgetting, we continue to train on *Group 1* in conjunction with a new set of tasks, *Group 2*, which is sampled with $\lambda =$

0.8. *Group 2*, described further below, contains tasks aimed at teaching more question-answering specific reasoning skills, with a bias towards RC datasets.

Our purpose in having two groups is to enable us to implement differing sampling strategies within a single training regime. For *Group 1* we utilise uniform sampling over all tasks and for *Group 2* we use error-based sampling. This combination represents our solution to the issue noted in Yoran et al. (2022), namely that excessive oversampling will occur for tasks that the model cannot learn well. In addition we find uniform sampling useful for regulating the sampling of the tasks that the model has already learned in Stage 1.

### 2.4.3 *Base* and *Base+RATD* Models

We now discuss two resulting models, both continuing training from the best Stage 1 checkpoint and using the same *Group 1* tasks but different in *Group 2* tasks.

The first, our *Base* model, uses 41 tasks in *Group 2* for an overall total of 79 tasks (38 *Group 1* + 41 *Group 2*). *Group 2* consists of a diverse range of question-answering datasets. Of note, to facilitate an ability to identify relevant information and perform deductively valid reasoning, for HotpotQA, Hover, FEVER, Musique, Natural Questions, CREAK (Onoe et al., 2021) and TriviaQA (Joshi et al., 2017), we construct fully evidential contexts with many irrelevant distractors using a combination of gold and distractor paragraph fragments such that we are as close to our maximum sequence length of 512 tokens as possible without truncating sentences. Since some evaluation samples have a label of "unanswerable", we also create versions of HotpotQA, Hover, FEVER and Musique by similar construction to the fully evidential samples but with key gold sentences or paragraphs removed. These are assigned an "unanswerable" label.

For our second model, *Group 2* consists of the 41 tasks in the above *Base* Group 2 plus an additional 14 *RATD* datasets for a total of 55 tasks. Our resulting *Base+RATD* model is thus trained on a total of 93 tasks (38 *Group 1* + 55 *Group 2*). As described above, the *RATD* dataset contexts are constructed using our Iterator against the full Wikipedia corpus. Recalling that none of our original datasets are normalised against the version of Wikipedia we use, the resulting contexts are noisy, often containing partial or no relevant evidence and many distractors. We hypothesise that the utility of these is to impart a variety of heuristic strategies using a context form similar to that which our downstream unseen evaluation datasets will have. Thus our *Base+RATD* model may be equipped for reasoning to a plausible answer from partial information as well as the deductively valid answer derivable for the majority of datasets used to train the *Base* model.

Details of all datasets utilised in QA Model training are in Appendix E.

### 2.4.4 QA Model In-domain Evaluation

Table 2: Comparison of our QA Model performance to related pretraining methods in *finetuned* setting on DROP dev set and IIRC test set (F1). [a] Pi et al. (2022); [b] Yoran et al. (2022) trained without digit tokenisation; [c] from Trivedi et al. (2022b) wherein PReasM is retrained with digit tokenisation; [d] Trivedi et al. (2022b).

| Pretraining Regime | DROP | IIRC$_G$ | IIRC$_R$ |
|---|---|---|---|
| POET-SQL (BART)[a] | 82.2 | | |
| PReasM (T5-large)[b] | 72.3 | 75.0 | 45.1 |
| PReasM w/digit tok. (T5-large)[c] | 80.0 | 73.3 | 40.9 |
| PReasM + Teabreac (T5-large)[d] | 83.2 | 77.9 | 47.6 |
| Teabreac (T5-3B)[d] | **86.7** | 79.5 | 51.0 |
| Ours: *Base* (BART) | 79.2 | **80.2** | **53.6** |
| Ours: *Base+RATD* (BART) | 79.6 | 80.1 | 52.8 |

For comparison with related approaches, we fine-tune our models on DROP (Dua et al., 2019) and separately on IIRC$_G$ and IIRC$_R$ (Ferguson et al., 2020). IIRC$_G$ is an oracle setting, with context consisting of gold sentences and surrounding text. IIRC$_R$ has a retrieved context using respective retrieval methods from each

study. As shown in Table 2 we are competitive with other approaches in this in-domain setting: We are slightly behind on DROP compared to POET (Pi et al., 2022) and Teabreac (Trivedi et al., 2022b), however we are state of the art on IIRC$_G$ and IIRC$_R$.

## 3 Experiments

Our experiments are aimed at answering three main research questions:

**R1.** What is the impact of adding *RATD* datasets to the QA Model *Base* training regime?

**R2.** How effective is pretraining for numerical literacy in the unseen setting for smaller language models?

**R3.** What is the performance differential between our QA model with differing evaluation dataset context configurations and high-performing comparisons in a similar unseen setting?

For each evaluation dataset, where possible we report our results against other zero/few-shot work. If known, we also report the current state of the art. As applicable for each dataset we report results without retrieval, with our retrieval (denoted Dataset$_R$), and with gold context (denoted Dataset$_G$ or similar).

To facilitate comparison against prior work on DROP (Dua et al., 2019) and IIRC (Ferguson et al., 2020) we use the numeracy-focused F1 calculation introduced in Dua et al. (2019) whereby if the gold label is a number, the predicted answer must contain that number irrespective of other token overlap. For consistency we retain this method for reporting F1 for other datasets noting this is equivalent to standard F1 where the gold answer is not a number and disadvantageous to our results where the gold answer is a number. For datasets with binary labels we adopt the calculation used in Khashabi et al. (2020b) where to count as a match the predicted answer must appear in the gold label and the opposing answer must not. For multi-choice evaluation, we take the option with the highest overlap with the predicted answer and then score as exact match.

Where comparing performance of our *Base* against *Base+RATD* models we use the paired bootstrap test (Efron & Tibshirani, 1993) to test for statistical significance ($p < 0.05$).

We report results against the dataset splits commonly reported by relevant prior work.

### 3.1 Unseen Evaluation Datasets

**StrategyQA** (Geva et al., 2021) contains binary-labeled commonsense samples requiring a diversity of multi-hop reasoning strategies. The form of questions is generally implicit, meaning they do not leak information as to how they could be decomposed (e.g. "Did Aristotle use a laptop?" versus "Was Aristotle alive at the time that laptops were invented?"). Many samples involve reasoning to a plausible rather than an entailed conclusion even where gold paragraphs are provided (Liang et al., 2022) e.g. "Is greed the most prevalent of the Seven Deadly Sins?". To facilitate comparison with other zero-shot approaches we use the full training set for evaluation as per BIG-bench (Srivastava et al., 2022) (denoted SQA for question-only and SQA$_R$ for question plus our retrieval). We also report results with two forms of gold context; using the provided summary notes which have a short paragraph, rationale-like form (SQA$_{GF}$), and using the full paragraphs from each of three annotators (SQA$_{GP}$) - for brevity we report the mean score over the three gold paragraph sets.

**CommonsenseQA** (Talmor et al., 2019) (CSQA) is a 5-way multi-choice dataset of commonsense questions derived from Conceptnet (Speer et al., 2017). Many of the questions involve commonsense knowledge that is unlikely to be retrievable from a generic corpus ("Where on a river can you hold a cup upright to catch water on a sunny day"). However retrieving specific related examples such as "At the river, I filled my cup at a waterfall" may sometimes be possible (Piktus et al., 2021). CSQA augmented with our retrieval is denoted CSQA$_R$.

**DROP** (Dua et al., 2019) is a RC dataset wherein answering each question requires simple numerical or temporal reasoning. Questions only make sense with the provided gold paragraph so we do not perform retrieval. Answers may be numbers, dates or text spans.

**IIRC** (Ferguson et al., 2020) contains questions where an initial paragraph is given and answers depend upon this plus additional paragraphs that must be retrieved. Each sample is provided with links to all supporting documents, and prior work leverages these to restrict the number of documents to be retrieved from. We instead use our Iterator to augment samples from the full Wikipedia corpus using the concatenation of question and initial paragraph as the query, without reference to the given links ($IIRC_R$). We also report comparison against an oracle context ($IIRC_G$). Answers may be numbers, binary, text spans or labeled unanswerable. For $IIRC_G$ unanswerable samples, we construct contexts using the initial paragraph fragment plus 1-2 random distractor paragraphs.

**ARC-DA** (Bhakthavatsalam et al., 2021) is a question-only subset of ARC (Clark et al., 2018) where questions have been re-worded to make sense in an open domain context. The original multichoice versions of ARC are part of our training regime, hence compositionality is doubtful and samples are only partially unseen in the sense that the question format is different (and we use the test split). Nonetheless we report results in the interests of exploring diversity. We experiment with Iterator-augmented ($ARCDA_R$) versions as well as with a gold context that we construct from Worldtree (Xie et al., 2020) ($ARCDA_G$).

**Musique** (Trivedi et al., 2022a) is a multihop dataset constructed by combining single-hop questions from existing datasets including SQuAD (Rajpurkar et al., 2016) which is also part of our training regime. Moreover we utilise the training split of Musique in both our retriever and QA model training. However the provided development split has been constructed such that for all samples no single hop question, answer, or associated paragraph is common to the corresponding element of any training sample. Therefore we construct a new development set from the training set and experiment with the original Musique development split as "partially seen", this time where the form of questions is "seen" but the exact questions are not. Prior work generally uses specialised retrieval for Musique where selection is from the set of gold and distractor paragraphs provided for each sample. We experiment with our retrieval ($Musique_R$), and with a gold context constructed from gold paragraphs ($Musique_G$).

## 3.2 Models

The Retriever component of the Iterator is built upon RoBERTa-base (Liu et al., 2019) and both the Reranker and Evidence Set Scorer use ELECTRA-large (Clark et al., 2020). Unless noted otherwise, all results are reported against the same two final QA Models which are based on BART (Lewis et al., 2020). All models use the the Huggingface (Wolf et al., 2020) implementations.

## 3.3 Experimental Results

### 3.3.1 StrategyQA and CommonsenseQA

*Base+RATD* significantly outperforms *Base* on StrategyQA (Table 3). On $SQA_R$ (which uses our retrieved contexts) our much smaller *Base+RATD* model slightly exceeds performance of the two 11 billion parameter models and is comparable with OPT 175B (Zhang et al., 2022).

Our *Base* model fails to improve with $SQA_{GP}$ (which has contexts of gold paragraphs) versus the question-only SQA version. The improvement on $SQA_{GP}$ with the addition of *RATD* draws attention to the fact that outside of our *RATD* datasets the majority of our multihop training samples are aimed at imparting deductively valid forms of reasoning which, as noted above, are often inapplicable for $SQA_{GP}$.

As described in section 3.1, the contexts of $SQA_{GF}$ are of a condensed, rationale-like form, distinct from the standard verbose paragraph form of $SQA_{GP}$. Model performance on $SQA_{GF}$ hugely outperforms our other configurations. This shows that with a context of a form the model has learned to reason with, it is possible to solve challenging implicit questions. As to where our models may have learned to reason with this short context form we note that some of the training datasets contain similar short form contexts e.g. BoolQ (Clark et al., 2019b), which like StrategyQA has binary labels. Our *Base* model has 84.9% development set accuracy on BoolQ.

As Table 4 shows, augmenting CommonsenseQA samples with retrieval ($CSQA_R$) yields mixed results. Others e.g. Piktus et al. (2021) have observed that the best zero/few shot performance on this type of

Table 3: StrategyQA performance comparison (Accuracy). StrategyQA contains binary-labelled, multi-hop commonsense questions. Bold figures denote the better of our two models. All *Base* versus *Base+RATD* differences are statistically significant. [a] Wang et al. (2022a); [b] Tay et al. (2022); [c] Chowdhery et al. (2022); [d] from Taylor et al. (2022); [e] Sanh et al. (2021); [f] Khashabi et al. (2022) [g] Below-random performance on our *Base* model with Q+retrieval is due to the model predicting text other than yes or no. Prepending "Yes or no -" to each question improves the score from 48.4 to 54.9. The corresponding *Base+RATD* figure is 58.8 which retains statistical significance.

| Model | Params | Base | Base+RATD |
|---|---|---|---|
| Random | | 50.0 | 50.0 |
| PaLM - COT+Self-cons.[a] | 540B | 81.6 | |
| U-PaLM - 5 shot[b] | 540B | 78.3 | |
| PaLM - 5 shot[c] | 540B | 73.9 | |
| OPT - 5 shot[d] | 175B | 58.5 | |
| T0++[e] | 11B | 54.4 | |
| UnifiedQA v2[f] | 11B | 57.9 | |
| PaLM - 5 shot | 8B | 55.4 | |
| UnifiedQA v2 | 770M | 51.6 | |
| Ours: SQA | 440M | 51.6 | **53.9** |
| Ours: SQA$_R$ (Our retrieval) | 440M | 48.4[g] | **58.9** |
| Ours: SQA$_{GF}$ (Gold facts) | 440M | **72.8** | 71.2 |
| Ours: SQA$_{GP}$ (Gold paras) | 440M | 51.6 | **55.8** |

Table 4: CommonsenseQA development set performance comparison (Accuracy). CommonsenseQA contains multi-choice commonsense questions. Bold figures denote the better of our two models. *Base+RATD* improvement is statistically significant for CSQA but not for CSQA$_R$ (adding retrieved context improves *Base* but not *Base+RATD*). [a] Xu et al. (2021); [b] Chowdhery et al. (2022); [c] Brown et al. (2020); [d] Khashabi et al. (2020b)

| Model | Params | Base | Base+RATD |
|---|---|---|---|
| Random | | 20.0 | 20.0 |
| Prior work (finetuned)[a] | 418M | 91.2 | |
| PaLM - 0/5 shot[b] | 540B | 69.2/81.5 | |
| GPT3 - 0/few shot[c] | 175B | 81.5/85.0 | |
| UnifiedQA v1[d] | 11B | 76.2 | |
| PaLM - 0/5 shot | 8B | 66.0/77.6 | |
| GPT3 - 0/few shot | 760M | 61.8/62.7 | |
| UnifiedQA v1 | 770M | 60.9 | |
| Ours: CSQA | 440M | 61.1 | **64.0** |
| Ours: CSQA$_R$ (Our retrieval) | 440M | 62.4 | **63.6** |

dataset has been achieved with much larger models rather than external retrieval and our analysis bears this out.

The addition of extra reasoning strategies via the *RATD* datasets is more successful; as with StrategyQA, performance on CommonsenseQA is improved with the *Base+RATD* model.

### 3.3.2 DROP and IIRC

As with PaLM, our *Base* and *Base+RATD* models are trained using digit tokenization. On DROP both our models outperform all models not trained using this method including GPT3 175B and InstructGPT 175B (Ouyang et al., 2022) (Table 5). Performance of our models approaches that of PaLM 8B and PaLM 540B in the zero shot setting but both are superior to ours with a 5-shot prompt.

Table 5: DROP development set performance comparison (F1). DROP primarily tests numeracy in reading comprehension. Reduced performance on *Base+RATD* versus *Base* is statistically significant. [a]Chowdhery et al. (2022); [b]Brown et al. (2020); [c]Ouyang et al. (2022); [d] Khashabi et al. (2020b)

| Model | Params | Base | Base+RATD |
|---|---|---|---|
| PaLM - 0/5 shot[a] | 540B | 43.7/70.8 | |
| GPT3 - 0/few shot[b] | 175B | 23.6/36.5 | |
| InstructGPT PPO+ptx - 0/few shot[c] | 175B | 15.2/33.3 | |
| UnifiedQA v1[d] | 11B | 32.5 | |
| PaLM - 0/5 shot | 8B | 45.1/69.4 | |
| UnifiedQA v1 | 770M | 24.6 | |
| GPT3 - 0/few shot | 760M | 14.4/24.0 | |
| Ours | 440M | **40.7** | 40.0 |

Ablative experiments on our training regime components (Table 6) indicate that digit tokenization, numerical literacy training datasets and two stage training are all important in achieving the best DROP performance in our setting.

Table 6: DROP development set (F1). Ablative results on our QA models trained using *Base+RATD* datasets trained in one or two stages, with/without digit tokenization (+/-DT), and with/without numerical literacy training datasets (+/-NumLit). Note that the -NumLit setting is only relevant for single-stage training.

| Model | All Ans. Types | Numeric Ans. Only |
|---|---|---|
| Two Stage: +DT +NumLit | **40.0** | **25.4** |
| One Stage: +DT +NumLit | 38.2 | 22.9 |
| Two Stage: -DT +NumLit | 34.7 | 16.6 |
| One Stage: +DT -NumLit | 29.0 | 11.2 |

Table 7: IIRC test set evaluation (F1). IIRC tests diverse reasoning requiring retrieval. Both *Base* to *Base+RATD* comparisons are statistically significant. [a] Ferguson et al. (2022) use a finetuned QA model and specialised retrieval with corpus restricted to documents linked from each initial paragraph. [b] To the best of our knowledge our *Base* model finetuned on IIRC$_R$ and separately on IIRC$_G$ are both SOTA at the time of writing so we report these given unavailability of unseen comparisons.

| Model | Params | Base | Base+RATD |
|---|---|---|---|
| Prior work: IIRC$_R$[a] | 123M | 51.6 | |
| Ours: Finetuned IIRC$_R$ (Our retrieval)[b] | 440M | 53.6 | |
| Ours: IIRC$_R$ (Our retrieval) | 440M | 23.8 | **25.5** |
| Ours: Finetuned IIRC$_G$ (Gold context)[b] | 440M | 80.2 | |
| Ours: IIRC$_G$ (Gold context) | 440M | **59.6** | 58.1 |

Table 7 shows performance on IIRC. A first glance suggests that poor retrieval is the major cause of low performance on IIRC$_R$, however inspection of retrieved items suggests that retrieval is often fully evidential. The breakdown by answer types in Table 8 indicates that a major cause of failure is that in contrast to DROP, almost all numeric answers are predicted incorrectly for both IIRC$_G$ (gold contexts) and IIRC$_R$ (retrieved contexts). Finetuning alleviates the issue, confirming that the model is capable of performing the necessary computation when trained with sufficiently similar examples.

Our *Base+RATD* model generally correctly predicts unanswerability for IIRC$_G$ but almost never does for IIRC$_R$. The IIRC$_R$ context frequently contains either enough information to make the question answerable,

Table 8: Breakdown by answer type on DROP development set and IIRC test set (F1). Sample counts are in brackets. Finetuned models are trained from the *Base+RATD* checkpoint.

| Dataset | Ans. Type | Base+RATD | Finetuned |
|---|---|---|---|
| DROP | Span (2962) | 67.4 | 82.3 |
| | Multi-span (567) | 42.0 | 72.2 |
| | Num (5850) | 25.4 | 79.0 |
| | Date (157) | 62.4 | 74.0 |
| | All (9536) | 40.0 | 79.6 |
| IIRC$_G$ | Span (544) | 59.8 | 74.3 |
| | Binary (66) | 57.1 | 64.7 |
| | Num (277) | 2.9 | 67.4 |
| | No answer (414) | 92.8 | 98.8 |
| | All (1301) | 58.1 | 80.1 |
| IIRC$_R$ | Span (544) | 48.9 | 44.8 |
| | Binary (66) | 68.2 | 57.6 |
| | Num (277) | 3.8 | 41.5 |
| | No answer (414) | 2.6 | 69.9 |
| | All (1301) | 25.5 | 52.8 |

or more frequently such relevant information as to make it *appear* answerable. Similar to the numerical computation issue, adding sufficiently similar training examples via finetuning enables the model to distinguish unanswerable samples. Appendix G illustrates failure cases for numeric and unanswerable types.

### 3.3.3 ARC-DA and Musique

Table 9: ARC-DA (test accuracy) and Musique (development F1) comparisons. ARC-DA is science question answering and Musique involves multi-hop question answering. All *Base* to *Base+RATD* differences are statistically significant. Musique performance degradation in *Base+RATD* is caused by *adding* Musique *RATD* in training; results for an ablative model trained with all datasets *except* for Musique *RATD* is shown in brackets. [a] Bhakthavatsalam et al. (2021): Training includes ARC-DA. [b] Trivedi et al. (2022a): EX(SA) uses specialised retrieval from each Musique sample's gold and distractor paragraphs.

| Model | Params | Base | Base+RATD |
|---|---|---|---|
| UnifiedQA+ARCDA/MC with IR[a] | 11B | 61.4 | |
| Ours: ARCDA$_R$ (Our retrieval) | 440M | 28.8 | **31.6** |
| Ours: ARCDA$_G$ (Gold context) | 440M | 56.8 | **59.1** |
| Musique - EX(SA)[b] | 102M | 49.8 | |
| Ours: Musique$_R$ (Our retrieval) | 440M | 24.3 | 22.2 (28.2) |
| Ours: Musique$_G$ (Gold context) | 440M | 60.8 | 43.8 (62.4) |

Table 9 shows model performance on our "partially seen" datasets, ARC-DA and Musique. On ARC-DA, adding *RATD* datasets significantly improves results in both retrieved and gold settings. By contrast, Musique performance significantly degrades with *Base+RATD*. Noting that Musique is the only evaluation dataset for which we create *RATD* datasets, we hypothesise that in the case of highly similar training examples to particular evaluation samples, the model prediction is the memorised answer of a similar training example. We confirm this by examining the predicted answers of the 1,670 Musique evaluation samples that scored 0 F1 against *Base+RATD*. Of these the predicted answers of 716 samples are an exact match to a Musique training sample gold answer (e.g. "Who is the spouse of the Green performer?" is incorrectly answered as "anna gordy gaye" because this is the label to a number of training questions of "Who is the

spouse of ..." form). An ablative experiment, wherein we trained a version of *Base+RATD* without the Musique *RATD* datasets, results in improved performance versus *Base* and the original *Base+RATD* on Musique (Table 9) without material impact to other evaluation dataset results.

The Musique training split has 19,938 samples but only 2,057 unique labels, and questions with the same answer tend to be of similar form, such as the above "Who is the spouse of..." example. Therefore we consider the question of whether the poor performance of *Base+RATD* here is a general weakness of our method or whether it is specific to the particular bias of Musique. We trained another *Base+RATD* model, this time with the Musique *RATD* training dataset substituted with a filtered variation that only contains samples with unique labels. Similar to the above Musique *RATD* ablation, this version also improves against the original *Base+RATD* (+3.0 F1 for Musique$_R$ and +10.6 F1 for Musique$_G$) without impact to other results. Hence, assuming appropriate consideration of existing dataset bias when selecting *RATD* training samples, we affirm the robustness of our method.

## 4 Conclusion

We have argued that an ability to reason over imperfect and incomplete information is a critical skill with which question-answering models must be endowed. To facilitate such ability we create *RATD* datasets that are designed to impart heuristic reasoning strategies with context of a form similar to that which retrieved contexts for downstream tasks will have. We show that training on *RATD* datasets improves performance on all unseen evaluation datasets with retrieved contexts. This sometimes comes at a small cost in situations where questions come with gold contexts that are in a form that our model is already good at utilizing (SQA$_{GF}$, DROP, and IIRC$_G$) although we suggest that in practice such gold contexts are the less common case. **(R1)**

We also show that even with our large and diverse pre-training regime, questions involving numerical computation and those labelled unanswerable remain sensitive to the similarity of training samples. **(R2)**

Our results demonstrate that generic retrieval without normalisation can outperform specialised methods (e.g. we are state of the art on fine-tuned IIRC$_R$) and that our overall method can yield performance on par or better than that of much larger models without fine-tuning (e.g. SQA$_R$, DROP). **(R3)**

### Broader Impact Statement

In common with the well known issues with larger models, our system is capable of generating hallucinated, false and/or potentially offensive answers. We consider its usage at this stage of development to be most appropriate in research environments.

Conversely, latency, physical compute size, cost and energy efficiency are important considerations where smaller models offer material benefits. As noted we suggest that a diversity of applications exist in the broad domain of reasoning systems and that due weight be assigned to all factors in determining the most appropriate approach for a given situation.

### Acknowledgments

We are grateful to the authors of Pi et al. (2022) for providing us with their POET-SQL dataset and to Omar Khattab for providing us with Hover paragraph sequencing data.

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

## A  Hyperparameters

All models are trained on one GPU (either an Nvidia RTX8000 or A100) except for the Retriever models which are trained on six 80GB A100 GPUs. All models are trained using mixed precision using a linear learning rate decay schedule. Initial learning rates and other hyperparameters are shown in Table 10. The optimiser used for the Retriever, Reranker and Evidence Set Scorer is Adam. All other models use AdamW. A maximum sequence length of 512 tokens was used for all models.

Table 10: Hyperparameters used for each model. Each training step is one batch input i.e the number of optimization steps is $TrainingSteps/GradientAccumulationSteps$. All final models are selected as the best model on the development set(s) over the specified number of training steps.

| Model | Initial LR | Batch Size | Grad. Accum | Train Steps |
|---|---|---|---|---|
| Retriever | 2e-5 | 150 | 1 | 99K |
| Retriever+memory bank | 1e-5 | 250 | 1 | 59K |
| Paragraph Reranker | 5e-5 | 12 | 8 | 140K |
| Evidence Set Scorer | 5e-5 | 12 | 8 | 140K |
| QA Model Stage 1 | 2e-5 | 32 | 4 | 1M |
| QA Model Stage 2 *Base* | 2e-5 | 32 | 4 | 1M |
| QA Model Stage 2 *Base+RATD* | 2e-5 | 32 | 4 | 1M |
| DROP finetuned | 2e-5 | 32 | 4 | 260K |
| $IIRC_G$ finetuned | 2e-5 | 32 | 4 | 40K |
| $IIRC_R$ finetuned | 2e-5 | 32 | 4 | 40K |

## B  QA Model Input Format

We employed a simple and fixed input format based on that used in UnifiedQA (Khashabi et al., 2020b) with minor extensions as follows:

Open domain form:
```
question? \\n
```

Reading comprehension (RC) form:
```
question? \\n paragraph(s)
```

Multiple choice form:
```
question? \\n (A) option text a (B) option text b ...
```

Multiple choice with RC form:
```
question? \\n (A) option text a (B) option text b ... \\n paragraph(s)
```

We standardised the formatting of any paragraphs or paragraph fragments that had associated document titles as follows. Further detail on how such contexts were constructed is in Appendix E.

```
Title 1: Sentence 1. Sentence 2. Title 2: Sentence 1. Sentence 2. ...
```

## C  Wikipedia Corpora

For experiments aimed at evaluating the Iterator components in an in-domain setting (Table 1), we used the same corpus of Wikipedia abstracts from October 1 2017 that HotpotQA and Hover are based upon.

For our main experiments and for various peripheral tasks such as identifying negative paragraphs for retrieval training we start with the August 1 2020 Wikipedia dump as preprocessed by (Qi et al., 2021). We retain all paragraphs with more than seven words, and extract hyperlinks and calculate sentence offsets from each. There are a total of slightly over 35 million paragraphs. We note that all results in this paper use the original HotpotQA question set rather than the version used in (Qi et al., 2021) that has been normalised against this Wikipedia version.

## D    Iterator Training Details

### D.1    Retrieval Model Additional Details

Our final Retrieval model was trained similarly to Xiong et al. (2021) in that following the initial stage of training, additional training with a large memory bank (Wu et al., 2018) of negative paragraph embedding vectors was applied.

For retrieval of paragraphs for $RATD$ datasets, the number of paragraphs retrieved at each hop ($k$) was set to 60 so as to complete in reasonable time. In building unseen evaluation dataset contexts $k$ was arbitrarily set to 150 to maintain reasonable performance on queries that are very different to those used in retrieval training.

We used FAISS (Johnson et al., 2019) for the search over paragraph embedding vectors. Generally we used an approximate search mechanism, HNSW (Malkov & Yashunin, 2018), except for the Hover experiment (Table 1) where an exact inner product search was employed.

### D.2    Reranker Model

The Reranker has an input format as follows:

```
[CLS] query [SEP] yes no [INSUFF] [SEP] title [SM] sentence 0.  [SM] sentence 1.  ...
[SEP]
```

The query component is encoded as:

```
question [QSEP] title 1 | sentence 1.  sentence 2.  [QSEP] title 2 | sentence 1 ...
```

Special tokens are utilised as follows:
[CLS]: Trained using a one-layer head to be the Paragraph relevance score with a binary cross-entropy objective.
[INSUFF]: Insufficient Evidence token, used by the start and end token span predictors that are implemented as per Devlin et al. (2019). Although we utilise a separate abstractive QA model, we use the span predictors as a debugging tool and retain this component in the final loss function.
[SM]: Sentence Marker(s). Used to score sentence relevance. Trained using a one-layer head with a binary cross-entropy objective.
[QSEP]: query components separator.

The final training objective is the unweighted summation of the paragraph relevance loss, sentence relevance loss and span loss.

### D.3    Evidence Set Scoring Model

This model has an input format as follows:

```
[CLS] question [SEP] yes no [INSUFF] [SEP] [SM] title 1 | sentence 1.  [SM] title 1 |
sentence 2.  [SM] title 2 | sentence 1 ...  [SEP]
```

Special tokens are utilised as follows:

[CLS]: Evidence Set score. Trained using a one-layer head with binary cross-entropy. The label is 1.0 if all of the gold sentences from all gold paragraphs are present and zero otherwise.

[INSUFF]: Insufficient Evidence token, as per the Reranker model.

[SM]: Sentence Marker, as per the Reranker model.

The final training objective is the unweighted summation of the evidence set loss, sentence relevance loss and span loss.

Following Khattab et al. (2021), the maximum number of sentences in an evidence set was set to nine in all experiments. To select the sentences for constructing the retriever query and evidence set for the next hop a maximum of five sentences over a threshold are selected, also following Khattab et al. (2021). The minimum threshold used to select sentences is 0.1 unless fewer than 2 sentences qualify in which case the two top-scoring sentences are taken.

## E   QA Model Multitask Training Details

We trained both the first and the second stage for one million steps (batches) with the best model defined as that with highest mean exact match accuracy over all development sets. To ensure reasonable elapsed time for each validation step we used *reduced* development sets where development sets of more than 1250 samples were reduced to approximately 1250 by taking every $n$th sample with $n = round(c/1250)$ where $c$ is the sample count. A validation step occurs every 10,000 training steps.

Table 11 enumerates datasets used in Stage 1 and in Stage 2 Group 1 (those above the dotted line were added for Stage 2, namely CREAK (Onoe et al., 2021), CommonsenseQA 2.0 (Talmor et al., 2021), TriviaQA (Joshi et al., 2017), Natural Questions (Kwiatkowski et al., 2019) and Twenty Questions[3]). During Stage 1 training, error-based sampling for these datasets was employed and in Stage 2, uniform sampling.

Datasets names containing the term "opendomain" only use the question text as input and are added with the primary aim of teaching the model about the expected form of answer for a given question type (e.g. yes or no for "Could an Aardvark use a knife and fork?".

Datasets preceded by "preasm" are as provided by Yoran et al. (2022) with reformatting into our standard form. Datasets preceded by "poetsql" are the POET-SQL dataset kindly directly provided to us by the authors of Pi et al. (2022). We split POET-SQL into separate datasets based on the type of SQL statement and converted into our standard form.

For the "synthetic_num" datasets we extended the original code provided by Geva et al. (2020) to output in the variablised form proposed in (Pi et al., 2022) (e.g. "1 + 3" becomes "x + y \\n x=1; y=3; z=0; ..." where z is a distractor). Additionally we added two datasets with questions of the form "Is x > | < | between y [and z]?" for numbers and dates respectively. We generated one million samples for each of the resulting eight datasets. The "synthetic_textual" task is as provided by Geva et al. (2020) aside from reformatting into our standard format.

Finally, we created a self-supervised task (enwiki_20200801_selfsvised), by sequentially concatenating paragraphs from documents in our Wikipedia dump until a sequence length of approximately 512 tokens was reached. During training, spans were masked from each sample input based on their being named entities (Guu et al., 2020) or noun phrases with $\lambda = 0.65$, or randomly with $\lambda = 1 - 0.65$. The training objective was to predict just the masked spans as with T5 (Raffel et al., 2020) rather than the original BART (Lewis et al., 2020) objective of predicting the entire unmasked input sequence. A small development set was randomly selected to enable this task to be included with other tasks in model selection.

Table 12 enumerates datasets contained in Group 2 for Stage 2 training. We converted TAT-QA (Zhu et al., 2021) to our format by linearising the constituent tables. A number of the other datasets (i.e. those whose names do not contain key terms described below) are provided by (Khashabi et al., 2020b; 2022). These are:

---

[3]https://github.com/allenai/twentyquestions

Table 11: QA model Stage 1 and Stage 2 Group 1 training datasets. All figures are Exact Match on *reduced* development sets from the single overall best model without per-dataset finetuning. Datasets above the dotted line were added for Stage 2.

| Dataset | Base | Base+RATD |
|---|---|---|
| creak_opendomain | 76.6 | 76.1 |
| csqa2_opendomain | 49.4 | 51.9 |
| triviaqa_open_opendomain | 8.0 | 7.4 |
| naturalquestions_open_opendomain | 5.4 | 8.7 |
| twentyquestions_opendomain | 88.8 | 87.9 |
| preasm_arithmetic_addition | 99.6 | 99.8 |
| preasm_arithmetic_superlatives | 97.9 | 97.9 |
| preasm_composition | 93.4 | 93.7 |
| preasm_composition_2_hop | 93.5 | 93.7 |
| preasm_conjunction | 80.2 | 81.0 |
| preasm_counting | 96.6 | 96.5 |
| preasm_every_quantifier | 99.8 | 99.6 |
| preasm_most_quantifier | 99.8 | 99.7 |
| preasm_numeric_comparison_boolean | 99.9 | 99.8 |
| preasm_numeric_superlatives | 98.1 | 97.9 |
| preasm_only_quantifier | 99.4 | 99.4 |
| preasm_temporal_comparison | 93.7 | 93.0 |
| preasm_temporal_comparison_boolean | 99.8 | 99.7 |
| preasm_temporal_difference | 94.3 | 95.1 |
| preasm_temporal_superlatives | 97.5 | 97.1 |
| poetsql_multi | 36.2 | 34.5 |
| poetsql_select_abs | 84.2 | 94.0 |
| poetsql_select_arith | 89.7 | 85.1 |
| poetsql_select_count | 80.8 | 80.2 |
| poetsql_select_max | 79.6 | 75.7 |
| poetsql_select_min | 82.5 | 81.3 |
| poetsql_select_sum | 50.6 | 52.7 |
| poetsql_single | 79.4 | 79.0 |
| synthetic_num_arg_min_max | 100.0 | 100.0 |
| synthetic_num_date_diff | 82.6 | 82.7 |
| synthetic_num_date_min_max | 93.2 | 95.7 |
| synthetic_num_min_max_avg | 69.3 | 68.8 |
| synthetic_num_percent | 99.0 | 98.2 |
| synthetic_num_signed_arith | 76.1 | 78.6 |
| synthetic_num_yn_dates | 99.8 | 99.8 |
| synthetic_num_yn_nums | 100.0 | 100.0 |
| synthetic_textual | 92.4 | 92.4 |
| enwiki_20200801_selfsvised | 22.5 | 24.1 |

AdversarialQA (Bartolo et al., 2020), ARC (Clark et al., 2016; 2018), BoolQ (Clark et al., 2019a), BoolQ-NP (Khashabi et al., 2020a), MCTest (Richardson et al., 2013), the yes/no subset of MultiRC (Khashabi et al., 2018), NarrativeQA (Kočiský et al., 2018), NewsQA (Trischler et al., 2017), OpenbookQA (Mihaylov et al., 2018), PhysicalIQA (Bisk et al., 2020), PubmedQA (Jin et al., 2019), QAConv (Wu et al., 2022), QASC (Khot et al., 2020), Quail (Rogers et al., 2020), Quoref (Dasigi et al., 2021), RACE (Lai et al., 2017), Reclor (Yu et al., 2020), Record (Zhang et al., 2018), Ropes (Lin et al., 2019), SocialIQA (Sap et al., 2019), SQuAD 1.1 (Rajpurkar et al., 2016), SQuAD 2 (Rajpurkar et al., 2018), TweetQA (Xiong et al., 2019) and Winogrande (Sakaguchi et al., 2020). For readability, we omit citations for other datasets already referenced.

Table 12: QA model Group 2 training datasets. All figures are Exact Match on *reduced* development sets from the single overall best model without per-dataset finetuning.

| Dataset | Base | Base+RATD |
|---|---|---|
| adversarialqa_all | 46.0 | 47.6 |
| ai2_science_middle | 67.2 | 63.2 |
| ai2_science_elementary | 67.5 | 69.9 |
| arc_hard | 56.5 | 54.2 |
| arc_hard_with_ir | 59.5 | 59.5 |
| arc_easy | 68.3 | 70.4 |
| arc_easy_with_ir | 77.5 | 79.3 |
| boolq | 84.7 | 84.2 |
| boolq_np | 82.0 | 81.7 |
| creak_goldplusdistractors | 85.2 | 83.8 |
| creak_ratd | | 85.9 |
| creak_ratd_max4paras | | 85.6 |
| csqa2_ratd | | 56.7 |
| csqa2_ratd_max4paras | | 57.7 |
| fever_goldplusdistractors | 85.9 | 89.2 |
| hotpotqa_fever_hover_noanswer | 83.5 | 76.9 |
| hotpotqa_goldplusdistractors | 65.9 | 66.7 |
| hotpotqa_ratd | | 53.0 |
| hotpotqa_ratd_max4paras | | 52.5 |
| hover_goldplusdistractors | 84.0 | 82.2 |
| hover_ratd | | 78.5 |
| hover_ratd_max4paras | | 77.2 |
| mctest | 91.3 | 90.0 |
| multirc | 100.0 | 100.0 |
| musique_goldplusdistractors | 88.0 | 87.2 |
| musique_noanswer | 96.6 | 95.6 |
| musique_ratd | | 74.4 |
| musique_ratd_max4paras | | 75.1 |
| narrativeqa | 30.0 | 29.1 |
| naturalquestions_goldplusdistractors | 56.5 | 58.8 |
| naturalquestions_open_ratd | | 40.9 |
| naturalquestions_open_ratd_max4paras | | 39.9 |
| newsqa | 44.3 | 44.4 |
| openbookqa | 67.2 | 69.2 |
| openbookqa_with_ir | 68.4 | 70.6 |
| physical_iqa | 66.9 | 67.0 |
| pubmedqa_pqal_short_ans | 99.2 | 100.0 |
| qaconv | 54.3 | 54.8 |
| qasc | 53.4 | 55.5 |
| qasc_with_ir | 72.0 | 70.8 |
| qasc_ratd | | 61.9 |
| qasc_ratd_max4paras | | 62.6 |
| quail | 78.1 | 76.1 |
| quoref | 71.5 | 70.2 |
| race | 76.4 | 74.8 |
| reclor | 43.0 | 41.4 |
| record | 53.1 | 53.1 |
| ropes | 77.6 | 81.8 |
| social_iqa | 75.1 | 74.0 |
| squad1_1 | 66.5 | 64.9 |
| squad2 | 66.2 | 67.4 |
| tatqa | 41.6 | 40.8 |
| triviaqa_goldplusdistractors | 63.9 | 65.3 |
| tweetqa | 34.5 | 33.6 |
| winogrande_xl | 69.7 | 69.1 |

Dataset names containing "ratd" are those created by us by concatenating the original question with the retrieved context from our Iterator as described in the main text.

Dataset names additionally containing the term "max4paras" use these same contexts but are truncated to the top 4 retrieved paragraph fragments. We found that sometimes longer and sometimes shorter contexts provided better results and hence we added both forms to provide diversity in length.

Dataset names containing the phrase "with_ir" have retrieved contexts provided by Khashabi et al. (2020b) which we use unmodified.

Contexts for dataset names incorporating the term "goldplusdistractors" are constructed using the positive and negative paragraphs from corresponding retrieval training datasets. In both cases the document title was randomly withheld ($\lambda = 0.1$). For positive paragraphs we included the gold sentences plus random other sentences if sentence-level annotation was available, otherwise the full paragraph text. For negatives we similarly included either random sentences or full text such that the length distribution of positive and negative paragraphs was similar.

Squad 2 provides some unanswerable training samples. We supplemented these by creating unanswerable samples from HotpotQA, Hover and FEVER positives in a similar manner to the "goldplusdistractors" datasets except here we randomly drop gold sentence(s) and/or full gold paragraphs such that there is guaranteed to be at least one missing gold sentence. We performed the same activity for Musique at the paragraph level. All unanswerable samples have the label string "<No Answer>".

## F   Paired Bootstrap P-values

P-values for all *Base* to *Base+RATD* model comparisons under the Paired Bootstrap test are in Table 13.

Table 13: Paired Bootstrap p-values. $SQA_{GP_x}$ denotes gold paragraphs from each of three annotators.

| Dataset | P-value |
| --- | --- |
| SQA | 0.008 |
| $SQA_R$ | 0.000 |
| $SQA_R$ w/ Yes or no prefix | 0.000 |
| $SQA_{GF}$ | 0.031 |
| $SQA_{GP1}$ | 0.000 |
| $SQA_{GP2}$ | 0.000 |
| $SQA_{GP3}$ | 0.000 |
| CSQA | 0.006 |
| $CSQA_R$ | 0.155 |
| DROP | 0.017 |
| $IIRC_R$ | 0.017 |
| $IIRC_G$ | 0.049 |
| $ARCDA_R$ | 0.001 |
| $ARCDA_G$ | 0.013 |
| $Musique_R$ | 0.001 |
| $Musique_R$ w/o Musique *RATD* | 0.000 |
| $Musique_G$ | 0.000 |
| $Musique_G$ w/o Musique *RATD* | 0.009 |

## G   Example Failure cases

Table 14 contains examples of failure cases for samples with numeric and "unanswerable" labels from the IIRC$_R$ test split.

Table 14: Example failure cases for IIRC$_R$ samples on the *Base+RATD* model. The top two rows have numeric labels, the bottom two are labelled unanswerable. Bolded context text highlights information that could be used in deriving an answer.

| Question / Answer | Retrieved Context (condensed) |
|---|---|
| How old was the Grand Olympic Auditorium at the time of New Regime playing a landmark concert there? **Gold answer: 60. Predicted Answer: 1924** | New Regime (American band): ... That landmark concert was held at the Grand Olympic Auditorium on April 13, **1984** ... Grand Olympic Auditorium: ... The venue was built in **1924** . . . |
| How old was Messe when the First World War started? **Gold Answer 30. Predicted answer: 28**. | Giovanni Messe: Messe was born ... on **10 December 1883**. 20th-century events: The First World War ... started in 1914 and ended in 1918... Military career of Adolf Hitler: He was 25 years old in **August 1914**, when Austria-Hungary and the German Empire entered the First World War. |
| What albums were ranked higher than "It Takes a Nation of Millions to Hold Us Back" in Rolling Stone's the 500 Greatest Albums of All Time? **Gold answer: <no answer>. Predicted answer: the beatles**. | It Takes a Nation of Millions to Hold Us Back: ... In 2003, Rolling Stone ranked the album number **48** on its list of the 500 Greatest Albums of All Time... maintaining the rating in a 2012 revised list. Rolling Stone's 500 Greatest Albums of All Time: ... **topped** by the Beatles' 1967 album **"Sgt. Pepper's Lonely Hearts Club Band"**, with a top 10 that featured four entries from the Beatles (Nos. 1, 3, 5 and 10), two from Bob Dylan (No. 4 and 9), and one each from the Beach Boys (No. 2), Marvin Gaye (No. 6), the Rolling Stones (No. 7) and the Clash (No. 8). |
| In what direction does the Goulburn River flow to Sugarloaf Creek? **Gold answer: <no answer>. Predicted answer: north west**. | Charles Bonney: ... was the first to overland sheep, bringing some 10,000 ... to **Sugarloaf Creek, Victoria station a tributary of the Goulburn River**... Goulburn River: ... The river flows generally north, then west, then north, then west... |

