# OpenReview forum: "Teaching Smaller Language Models To Generalise To Unseen Compositional Questions"
_TMLR — Accepted by TMLR_

### Review · Reviewer_HiAz · 2023-04-23

**Summary Of Contributions:**

The authors investigate zero-shot generalization in question-answering but for smaller models, in particular, using retrieval from a corpus that doesn’t necessarily contain sufficient information to answer the question.

The paper offers baselines on smaller LMs reasoning abilities in the zero-shot setting. The paper shows RATD augmentation would lead to improved performance; numerical literacy and unanswerability question answering tasks are difficult to train for without similar training examples; improved retrieval approaches lead to improved performance.

More specifically, for retrieval, the authors expand multi-hop dense retrieval (MDR) from 2-hop max to 4-hop max. A chain is built e.g., q -> d1, (q, d1) -> d0, (q, d1, d0) -> d3 (where q stands for a question and d_i stands for the i-th document). The authors use clever tricks (partly based on Wikipedia) to obtain negatives for retriever training. The retriever is trained on four datasets: HotpotQA, Hover, Natural Questions, Musique.

Then, the reranking and evidence set scorers are trained. The reranker combines a paragraph relevance score and the individual sentence relevance score. The evidence set scorer combines paragraph relevance score and sentence relevance in the context of the evidence set. In-domain retrieval and reranking evaluation are done on the Hover dev set (number of retrieved paragraphs = 25). Performance is much better using the authors’ retriever compared to MDR retriever.

QA models are further pretrained using two stages. Stage 1 (33 tasks) deals with improving numerical literacy. Group 2 (38 tasks including the prior 33 tasks) is co-trained with Group 1; Group 2 aims to teach models about the expected form of answer for a given question type. Another difference is that Group 1 training use uniform sampling of tasks, and Group 2 uses error-based sampling of tasks.

Retriever is based on RoBERTa-base; reranker and evidence set scorer are based on ELECTRA-large; QA models are based on BART.

A few results are as follows. Base+RATD (the augmented data) > Base on StrategyQA (zero-shot), and the performance is better or comparable with some of the 11B models and OPT-175B, which seems impressive. SQA-GF (explained at the beginning of Section 3.1) outperforms other configurations of contexts in StrategyQA. The authors’ model’s DROP performance is better than many baselines, despite the model being much smaller and the setting is zero-shot (see Table 5). The model also achieves good performance on IIRC, ARC-DA, but shows significantly decreased performance on Musique and the authors have provided an explanation.


**Audience:**

Yes

**Claims And Evidence:**

Yes

**Requested Changes:**

See above.

**Strengths And Weaknesses:**

Personally I have learned a lot from the related work section.

The few-shot performances of big LMs are necessary and it’s great that the authors have included them.

The motivation of having smaller models that specialize in question answering is great, and the performance is quite impressive given the comparison with large LMs (sometimes even much better than large LMs in the few-shot setting).

---

It would be prudent to qualify the tasks which the authors’ approach/model would excel in.
- Would the authors’ model work on any question answering task? How about long-context question answering tasks (e.g., answering questions about novels)?
- Is the approach tailored to open-domain question answering tasks or other types of tasks?

Explain paragraph relevance score and sentence relevance scores.
- Would the tradeoff in coefficients impact final reranker performance? Are the two scores on the same scale, and do they need to be tuned?
- The paper claims that the paragraph relevance score is often more accurate than the sentence relevance scores; is there empirical evidence for this claim?
- Similar issue for the evidence set scorer coefficients in the following paragraph.

Confusion on the number of tasks: I’m confused about how many tasks are in Group 1 and Group 2. Section 2.4.2 says there are 38 tasks in Group 2. Section 2.4.3 says there are 41 tasks in Group 2 (and 41+14 tasks for the second model). How did you obtain “total of 93 tasks” – is it 38+41+14? I think the authors could be more clear on what each cluster of tasks represent, and why the tasks are separated this way.

Below contains some concerns on experiments:

On StrategyQA experiments: “Our SQA-GF setting hugely outperforms our other configurations. This shows that with a context of a form the model has learned to reason with, it is possible to solve challenging implicit questions.”
- What specifically is the “form the model has learned to reason with” and are there any summary statistics on the “form” of learning?
- What’s an “implicit question” – definitions and examples would be helpful.

It’s unclear what the source of improvement is, for the DROP task. Is it because of digit tokenization, or any particular stage in the training pipeline?

It’s still unclear to me when RATD would be harmful. If the task data is too similar to RATD data, then would there be significantly worse performance as in Section 3.3.3?

Minor (feel free to ignore if it would take lots of effort): The GPT-3 performances are from the original GPT-3 paper, not updated GPT-3 models (with instruction tuning, etc.) in the past three years. If it’s easy to get those results, I would include them in Table 5. If not, then it’s fine.

---

### Review · Reviewer_bda2 · 2023-05-04

**Summary Of Contributions:**

This paper works in the context of question answering augmented by retrieval methods (i.e., retrieving relevant information in order to help answer the question, rather than storing all knowledge in LM parameters). The authors propose a new method that involves adding “retrieval augmented training data” (RATD) during training - that is, adding training datasets augmented with retrieved passages. One advantage that is proposed for this approach is that it can teach models to use heuristics to find plausible answers in cases where there is no answer that is perfectly entailed by the available material. The authors then perform an extensive set of experiments testing this approach; it is extensive both in the number of datasets that they consider and in the thoroughness with which they compare against baselines and prior literature. The RATD augmentation is in general a useful modification: in some cases it allows them to get a new SOTA or to create models that are not SOTA but are competitive with much larger models from prior literature. Finally, the authors also include some useful analyses of 2 major areas of weakness for models, namely numerical computation (where they show that additional finetuning is very helpful) and recognizing unanswerability.

**Audience:**

Yes

**Claims And Evidence:**

Yes

**Requested Changes:**

None of these changes are crucial for securing my recommendation; they are just things that I think would help the reader to keep track of important points in the discussion.
- RC1: In the table captions, give a brief one-phrase reminder of what the dataset is like (e.g, “Results on StrategyQA, a binary dataset of multihop commonsense reasoning questions”)
- RC2: In both the text and the tables, whenever using a subscript abbreviation, it would be helpful to remind what the subscript stands for, as I found it hard to keep track of this - e.g., SQA_GF, IIRC_G. I know that this might be a lot, so there’s definitely no need to include it everywhere, but the most helpful places to add a reminder would be where you discuss some comparison that depends crucially on the subscript - e.g., “Our Base model fails to improve with SQA_GP (when the question is augmented with paragraphs from annotators) versus the question-only SQA version”.
- RC3: To help contextualize the numbers in the tables, it might be helpful to list a chance baseline (50% for StrategyQA, 20% for CommonsenseQA, etc.)


**Strengths And Weaknesses:**

Strengths:
- S1: The overall framing seems likely to be useful for future work, as it has interesting thoughts about designing systems that prioritize different factors (e.g., compute efficiency vs. accuracy) and also about being able to reason from imperfect or incomplete information.
- S2: The proposed approach is well-motivated (both the RATD component and the other components). Though there are several components to the approach, the authors have given clear reasons for why each one is set up the way it is.
- S3: Empirically, the proposed approach is effective in many cases, and the authors are open about cases where it is not so helpful.
- S4: The experiments are extensive, covering a large range of datasets and a large number of baselines and comparisons to prior work.

Weaknesses: I did not identify any serious weaknesses. The main shortcomings were the following, but there is nothing wrong with these issues - they are just points that might limit the impact of the work, rather than problems that need to be fixed:
- W1: There is a lot going on in the paper, and sometimes I lost track of what each term referred to or what was being tested. See “Requested changes” below for some thoughts on quick things that could help with this.
- W2: The benefits provided by RATD are inconsistent: That is, in some cases it does not help, and in other cases it helps by only a small margin. However, I do not view this as a serious weakness: First, the idea is an interesting one, so it is useful to see what its effect is, whether positive or negative. Second, it is hard to get any improvement when working with messy large-scale datasets such as these, so the fact that any improvements were observed is impressive.

---

### Review · Reviewer_xHAu · 2023-05-29

**Summary Of Contributions:**


1. The authors proposed a method for improving the reasoning abilities of small language models by training them on Retrieval-Augmented Training Datasets (RATD).
2. A comprehensive set of baselines is provided for evaluating the zero-shot reasoning abilities of small language models.
3. The paper demonstrates how training on RATD datasets can improve the performance of small language models on unseen evaluation datasets such as StrategyQA, CommonsenseQA, etc.
4. The authors identified the brittleness of training for numerical literacy and unanswerability in the unseen setting and proposed effective extensions to the retrieval approach.

**Audience:**

Yes

**Broader Impact Concerns:**

It has a section for this.

**Claims And Evidence:**

Yes

**Requested Changes:**

- Presentation: Use more illustrative examples to clarify the pipeline and highlight the motivation for each component. Including clearer explanations and justifications for each design choice would help readers better understand the authors' decisions.
- Problem Formulation: The paper should better define the problem and the concept of compositionality.
- Evaluation: Authors could improve the experimental section by stating research questions first and then presenting detailed analyses and results. The current structure is somewhat difficult to follow. Additionally, the inclusion of more retrieval and reranking methods as baseline approaches would strengthen the evaluation.
- Related Work Discussion: Authors should consider adding more discussion about related work in the same research direction.

**Strengths And Weaknesses:**

### Strengths:

1. The paper addresses a significant problem by focusing on the reasoning capabilities of smaller language models while taking practical limitations, such as latency, cost, energy efficiency, and physical compute size, into account.
2. The introduction of Retrieval-Augmented Training Datasets (RATD) and the subsequent evaluation of their impact on improving the reasoning performances of small language models is an innovative contribution.
3. The paper features a comprehensive evaluation and comparison with prior research, encompassing both fine-tuned and zero-shot settings.

### Weaknesses:

1. The paper lacks sufficient detail regarding the construction of RATD datasets and how these datasets provide superior reasoning capabilities compared to existing datasets. An illustration example would help clarify how the pipeline works.
2. The definition of compositionality is not well-explained in the paper. Clarification is needed concerning the compositionality of questions within evaluation datasets, and the justification for the number of hops needed per example. Furthermore, a clearer definition of "seen" and "unseen" questions and their relation to the evaluation strategy would be beneficial.
3. The authors do not provide sufficient explanation about the choice of evaluation datasets, the configurations for each component depicted in Figure 1, and the selection of hyperparameters (such as z and t_max). An evaluation of each component's performance contribution would also be useful.
4. The paper's novelty is somewhat limited. Retrieval, reranking, and iterative learning for generalization is a common pipeline for modern QA systems, and the proposed method's specific advantages compared to existing pipelines are not clearly articulated.
5. The paper's presentation could be improved (see suggested changes below).

---

### Decision · Action_Editors · 2023-08-15

**Recommendation:** Accept as is

**Comment:**

The authors have made several changes already taking the reviewer suggestions into account. Upon further discussion, the reviewers all agree that this is a good paper with solid technical insights and experiments. However, a few minor concerns still remain around the writing and clarity of presentation. In particular, reviewers feel that the paper still lacks adequate motivation for the problem and that the writing can be improved in places.

Therefore, I'm recommending acceptance, and strongly urge the authors to make these minor revisions (as suggested by reviewer xHAu) for the camera ready version:
1. Begin the paper with more compelling examples rather than relying on numerous abstract sentences and ideas. For example, a single running example in the introduction can help the readers understand the problems and challenges more easily.
2. Better motivate and explain the key components of the framework. Again, an example or two may help the reader better understand the purpose of the different parts in the framework.

**Audience:**

This paper will inform future research into techniques that leverage retrieval and reranking for multi-task training of smaller QA models, and provides a framework for thinking about different trade-offs between accuracy, efficiency, model size, cost, etc. The RATD datasets and baselines reported in the paper will also serve as useful benchmarks.

**Claims And Evidence:**

This paper proposes a method for compositional question answering over unseen datasets, using a combination of retrieval, reranking and multi-task pre-training. The main claim is that training language models with multiple retrieval-augmented training datasets (RATD) allows for better zero-shot performance on unseen datasets, even with smaller model sizes (<500M parameters). The claims are supported with ample experiments, which also serve as good baseline numbers for further research in this direction.